

# Composition of terrestrial mammal assemblages and their habitat use in unflooded and flooded blackwater forests in the Central Amazon

André L. S. Gonçalves[1], Tadeu G. de Oliveira[2,3],
Alexander R. Arévalo-Sandi[1], Lucian V. Canto[1], Tsuneaki Yabe[4] and
Wilson R. Spironello[1]

[1] Grupo de Pesquisa de Mamíferos Amazônicos (GPMA), Instituto Nacional de Pesquisas da Amazonia (INPA), Manaus, Amazonas, Brazil
[2] Departamento de Biologia, Universidade Estadual do Maranhão, Cidade Universitária Paulo VI, CP 09, São Luis, Maranhão, Brazil
[3] Instituto Pró-Carnívoros, Atibaia, São Paulo, Brazil
[4] Hokkaido Research Center, Forestry and Forest Products Research Institute, Sapporo, Japan

Corresponding author
André L. S. Gonçalves,
sousa.alg@gmail.com

## ABSTRACT

Several forest types compose the apparently homogenous forest landscape of the lowland Amazon. The seasonally flooded forests (*igapós*) of the narrow floodplains of the blackwater rivers of the Amazon basin support their community of animals; however, these animals are required to adapt to survive in this environment. Furthermore, several taxa are an important source of seasonal resources for the animals in the adjacent unflooded forest (*terra firme*). During the low-water phase, the *igapó* becomes available to terrestrial species that make use of *terra firme* and *igapó* forests. Nonetheless, these lateral movements of terrestrial mammals between hydrologically distinct forest types remain poorly understood. This study tested the hypothesis that the attributes of the assemblages (abundance, richness, evenness, and functional groups) of the terrestrial mammals in both these forest types of the Cuieiras River basin, which is located in the Central Amazon, are distinct and arise from the ecological heterogeneity induced by seasonal floods. After a sampling effort of 10,743 camera trap days over four campaigns, two for the *terra firme* (6,013 trap days) and two for the *igapó* forests (4,730 trap days), a total of 31 mammal species (five were considered eventual) were recorded in both forest types. The species richness was similar in the *igapó* and *terra firme* forests, and the species abundance and biomass were greater in the *terra firme* forest, which were probably due to its higher primary productivity; whereas the evenness was increased in the *igapós* when compared to the *terra firme* forest. Although both forest types shared 84% of the species, generally a marked difference was observed in the composition of the terrestrial mammal species. These differences were associated with abundances of some specific functional groups, *i.e.*, frugivores/granivores. Within-group variation was explained by balanced variation in abundance and turnover, which the individuals of a given species at one site were substituted by an equivalent number of individuals of a different species at another site. However, the occupancy was similar between both forest types for some groups such as carnivores. These findings indicate that seasonal flooding is a relevant factor in structuring the composition of terrestrial

mammal assemblages between *terra firme* and floodplain forests, even in nutrient-poor habitats such as *igapós*. The results also highlight the importance of maintaining the mosaic of natural habitats on the scale of the entire landscape, with major drainage basins representing management units that provide sufficiently large areas to support a range of ecological processes (*e.g.*, nutrient transport, lateral movements and the persistence of apex predators).

## INTRODUCTION

With approximately 140 genera and 425 species, the Amazon harbors one of the richest mammalian faunas in the world (*Mittermeier et al., 2003*; *Jenkins, Pimm & Joppa, 2013*; *Quintela, Da Rosa & Feijó, 2020*). However, the number and composition of plant and animal species at any single location in the Amazon varies depending on the heterogeneity of the habitat, which is determined by different edaphic, climatic, topographic, hydrologic, and geologic conditions (*Zapata-Ríos et al., 2021*). In this vast biome, the vegetation type constitutes the primary biotic feature for understanding the spatial distribution of forest animals (*Borges-Matos et al., 2016*; *Galetti et al., 2009*).

Several forest types are found in the apparently homogenous forest landscape of lowland Amazonia (*Moraes et al., 2021*). Regionally, the pedological differences associated with the range of lentic and lotic environments form a mosaic of landscapes dominated by upland forests (hereafter referred to as "*terra firme*" forests). These are surrounded by diverse floodable habitats, which are alluvial forests and are locally known as *várzeas* and *igapós* depending on the origin of their waters (*Goulding, 1993*). *Várzeas* are nutrient-rich (whitewater) flooded forests, whereas *igapós* are nutrient-poor (clear and blackwater) flooded forests (*Junk & Piedade, 2010*). This contrast among environments often acts as a filter for the dispersion and establishment of species, thus playing an important role in shaping vertebrate assemblages (*Dambros et al., 2020*).

*Terra firme* forests lie above the maximum flood levels of lakes and rivers and account for more than 82% of the Amazon basin (*Melack & Hess, 2010*). In contrast, flooded forests (*várzeas* and *igapós*), which are situated on floodplains, cover approximately 17% of the basin (*Hess et al., 2015*; *Parrens et al., 2019*). Blackwater *igapós* (investigated here) only cover an area of about 180,000 km$^2$ (*Junk et al., 2015*; *Fassoni-Andrade et al., 2020*) and represent approximately 2.16% of the Amazon basin. Because blackwater rivers carry little suspended sediment, which is limited to deposition of very thin layers of fine sediments, their floodplains are narrow. Consequently, *igapós* are seldom more than 100–300 m in width and have low-fertility soils due to their acidic pH. Additionally, the abundance of trees and biomass is much lower in *igapós* than what is found in *terra firme* forests (*Wittmann & Junk, 2016*). Within *igapós*, the oscillating water levels are subjected to a predictable, long-lasting monomodal flood pulse of up to 10 m that lasts 280–290 days year$^{-1}$. This is sufficient to flood an entire *igapó*, leaving only the treetops above water
(*Ferreira et al., 2010*; *Junk et al., 2015*). These environmental differences lead to marked changes in the structure and floral and faunal composition of these forest types (*Haugaasen & Peres, 2005a*).

Inundation patterns in the Amazon have a strong influence on the structure of assemblages of birds (*Laranjeiras, Naka & Cohn-Haft, 2019*), trees (*Montero, Piedade & Wittmann, 2014*), ants (*Pringle et al., 2019*), primates (*Barnett & Jucá, 2018*), and bats (*Pereira et al., 2009*; *Bobrowiec et al., 2014*). As such, it is expected that the same effects apply to terrestrial mammals. Seasonal flooding is a limiting factor for terrestrial mammals since the available land area for foraging decreases during high-water phases. However, the forest mosaic created by inundation during the unflooded period may contribute to the persistence of species due to them having home ranges large enough for individuals to use resources as they become available in space and time (*Haugaasen & Peres, 2007*; *Parolin, Wittmann & Ferreira, 2013*). The different structures and compositions of Amazonian *igapó* and *terra firme* forests engender a spatiotemporal mosaic of resource availability that may result in landscape-scale seasonal movements of terrestrial vertebrates between these often neighboring forest types (*Dunning, Danielson & Pulliam, 1992*; *Hawes & Peres, 2014*). This reciprocal use of flooded and unflooded forests depends on the habitat availability and is governed by the flood pulse, and highlights the importance of complementarity habitats for terrestrial vertebrates in the Amazon.

The flood pulse mainly influences resource availability for fruit-and-seed-eating species. This availability is markedly seasonal with two annual peaks (*Parolin, Waldhoff & Piedade, 2010*). The first peak occurs at the highest water levels and this is when most *igapó* trees synchronously produce fruits that are consumed by aquatic organisms. The second peak happens at the lowest water levels, when many seeds and fruits that had been floating in the water lie exposed on the forest floor, thus allowing their consumption by terrestrial species (*Ferreira et al., 2010*; *Hawes & Peres, 2016*). Additionally, a higher abundance of insects occurs when the water level recedes (*Adis, 1984*). Moreover, the presence of carnivores, which are taking advantage of their prey's movement toward newly available environments, is increased (*Antunes et al., 2019*). This period of higher resource availability in *igapó* forests coincides with a period of reduced availability in the adjacent unflooded *terra firme* forests (*Haugaasen & Peres, 2005b*). As a result of this asynchrony, terrestrial mammals (*i.e.*, ungulates, rodents, carnivores, and insectivores) move annually into *igapó* forests to take advantage of these pulses of food resource availability (*Rocha, Ramalho & Magnusson, 2016*; *Antunes et al., 2019*).

Mammals are often considered good model organisms due to their relatively robust taxonomic resolution and diversity (*Schipper et al., 2008*). They play important roles in tropical forest ecosystems; particularly in trophic regulation as seed dispersers and predators, in nutrient cycling, carbon storage, and ultimately in the maintenance of forest structure (*Sinclair, 2003*; *Estes et al., 2011*; *Buendía et al., 2018*). Despite an increase in the studies related to the selection of habitats by mammals in the Amazon, there is still little information about this group, in particular ones that consider the complementarity between habitats. Some studies have found lateral seasonal movements of species between *várzeas* and contiguous *terra firme* forests and have shown that species migrate to *várzeas*

during the low-water season in the search for food, such as fruits, seeds, and shoots, and return to *terra firme* forests when inundation commences (*Salvador, Clavero & Pitman, 2011*; *Alvarenga et al., 2018*; *Costa, Peres & Abrahams, 2018*). Even though *igapó* forests are present at the edges of some of the largest Neotropical rivers, such as the blackwater Negro River and the clearwater Xingu and Tapajós Rivers (*Wittmann & Junk, 2016*), the movements of terrestrial mammals between *terra firme* forests and flooded blackwater forests has not yet been investigated.

The present study is the first comparison of the composition of the terrestrial assemblages in unflooded *terra firme* and adjacent seasonally flooded *igapó* forests located along a blackwater tributary of the Negro River during low-water phases of the flood pulse. Specifically, the study investigated (i) whether the structure of mammal assemblages in terms of species richness, evenness, relative abundance, and contribution to trophic guilds differed, and (ii) how assemblage composition varied among and within both forest types. Additionally, we compared the occupancy patterns of both forest types by each species. The lateral movement hypothesis was tested, which states that terrestrial mammals move seasonally from *terra firme* to *igapó* forests during low-water phases of the flood pulse. It was expected to find a similar mammal richness in the *igapó* forest and increased species abundance, biomass, and occupancy in *terra firme* forest, since *igapós* are poorer environments in terms of nutrients, with lower tree abundance and biomass reflecting on the vertebrate fauna that inhabits them (*Wittmann, Schöngart & Junk, 2010*). A greater or similar occupancy of *igapó* by shoot and seed-eating species as that of *terra firme* was also anticipated during the period in which *igapó* was available for terrestrial mammals and many seeds and fruits lie exposed on the forest floor (*Haugaasen & Peres, 2005b*). Moreover, it was predicted that shoot and seed-eating species greatly contributed to the abundance similarity between forest types as a result of the supply of these resources (*Parolin, Wittmann & Ferreira, 2013*; *Hawes & Peres, 2014*). Finally, the value of *igapó* forests as areas of special importance for the conservation of terrestrial mammals in the Amazon is discussed.

## STUDY AREA

The study area was in the Cuieiras River basin located in the Central Amazon (02° 37′ to 02° 38′ S, 60° 09′ to 60° 11′ W) (Fig. 1A). This basin is inserted in a mosaic that is composed of 12 Protected Areas on the Lower Negro River (*Didier et al., 2017*). It is an important area due to its diversity of species and habitat heterogeneity, as well as the presence of traditional communities. Specifically, the Cuieiras River basin contrasts with the nearby city of Manaus, the major city of the Amazon, which comprises more than two million urban inhabitants (approximately 60 km southeast) (*SEPLANCTI, 2016*) (Fig. 1B). The local population in the Cuieiras River basin is composed of six communities of indigenous and traditional peoples that inhabit the mouth of the river basin (*SEMA, 2019*). For subsistence and cash income, the residents are generally engaged in agricultural and extractive activities and ecotourism, which is of great importance for the region. Despite the presence of the communities, the forest retains a full complement system of vertebrate species (*Arévalo-Sandi et al., 2021*) and remains virtually undisturbed. It includes a unique

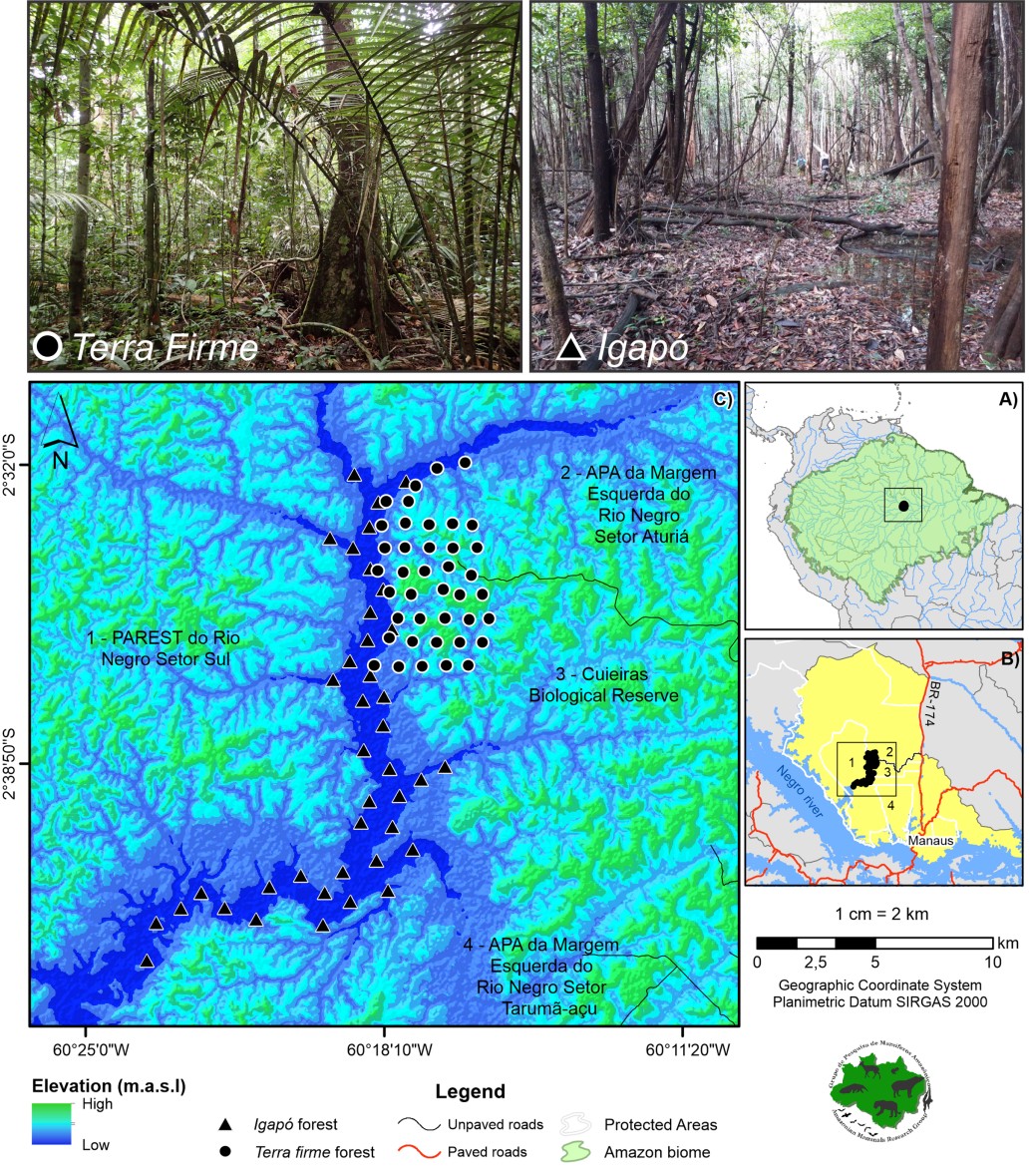

**Figure 1  Study area showing the camera trap points.** The cameras are distributed across the *terra firme* and *igapó* forests on the Cuieiras River basin, Central Amazon. Images of the unflooded forest (upper left) and the seasonally flooded forest during the low-water phase (upper right) are presented. Photo credit: André Gonçalves.

landscape mosaic that contains large expanses of *terra firme* (~86%) and *igapó* forest (~3%), the latter being seasonally flooded by the Negro River.

The *terra firme* forest habitats are unflooded forests with a relatively open understory and a dense uniform canopy, a height range of 30–39 m, and an emergent layer reaching 55 m in height. In these habitats, the soils are nutrient-poor, sandy and clayey Oxisols (*Gentry & Emmons, 1987*; *De Oliveira & Mori, 1999*). The average elevation range from 40 to 160 m above sea level with an undulating topography (*Prance, 1990*). These variations greatly affect the forest structure. Indeed, the forests have visibly different vegetation

**Table 1 Study area sampling effort.** Sampling area, type of habitat, date, and number of camera stations for the four camera trap surveys carried out at the Cuieiras River basin located in Central Amazon, Brazil. Camera*days are the number of days during which the survey was performed multiplied by the number of camera stations.

| Forest type | Sampling period (m/d/y) | Trap stations | Effort (camera*days) |
|---|---|---|---|
| *Terra firme* | 06/28/2018 to 10/15/2018 | 40 | 3,587 |
| | 07/15/2019 to 09/14/2019 | 40 | 2,426 |
| *Igapó* | 10/18/2018 to 01/05/2018 | 40 | 2,384 |
| | 10/19/2019 to 12/16/2019 | 40 | 2,346 |
| | | **160** | **10,743** |

formations that are associated with hilltops and slopes of varying inclinations. Additionally, bottomland swamps may occasionally occur in areas where plateaus are dissected by streams (*Baccaro et al., 2008*).

The *igapó* bathed by the Negro River is a seasonally flooded forest. The flooding by the blackwaters poor in nutrients and sediments (*Wittmann, Schöngart & Junk, 2010*) occurs on the banks of rivers and lasts for about nine months of the year. The flowering, fruiting, and leaf production of plants in the flooded environment show strong seasonal peaks (*Montero, Piedade & Wittmann, 2014*). Due to the lack of soil nutrients, tree abundance and biomass in *igapó* forests are much lower than those in *várzea* and *terra firme* forests (*Junk et al., 2015*). Tree species richness (≥10 cm diameter at breast height) of *igapó* forests ranges from nearly monodominant stands of shrub and tree species at deeply inundated sites to 20–35 species ha$^{-1}$ at mean inundation depths of 3–5 m. During the terrestrial phase, the herbaceous communities in the *igapó* are sparsely present at the edges of the forest and dominated by sedges, *e.g.*, *Rynchospora* spp. and *Scleria* spp. (*Wittmann & Junk, 2016*).

The average regional temperature is 26 °C, and the seasons are well defined. Most rainfall occurs from December to May (211–300 mm monthly average), and a marked dry season lasts from June to November (42–162 mm monthly average) (*Ribeiro & Adis, 1984*). In the flooded forests, floodwaters reach their peak in May–July, when they may be up to 6 m deep. The *igapó* is unflooded between late October and early February. Because of the sloping nature of the narrow *igapó* floodplain, some areas of the forest may be dry for less than four weeks a year, whereas other areas remain unflooded for 10–12 weeks (*Junk et al., 2011*).

## MATERIALS AND METHODS

### Data survey

For the study, two sampling campaigns, per forest type were conducted (Table 1). Forty camera traps (Reconyx, UltraFire XR6) were used in each forest type (*n* = 80) per year (Fig. 1C). During each field campaign, cameras remained at the sampling point for a minimum of 60 consecutive days (mean in days: 75.1 for the *terra firme* and 59.6 for the *igapó*), and the same original sampling locations were used for each campaign. The difference in the means is due to the fact that some cameras in the igapó stopped

working before the scheduled period, due to malfunctions, but this variation did not affect the sampling coverage. The cameras were deployed first on the *terra firme* and, soon after, were reallocated to the adjacent *igapó* forest. Based on Sentinel satellite images for the year 2018 and ALOS PALSAR radar images, the camera trap points were first defined for each forest type and then a field validation was performed. In the *terra firme* forest, the camera trap points were arranged on sets of trails, whereas they were installed along the extension of the basin in the *igapó* forest. The cameras were spaced 1 km apart in both environments.

All cameras were placed on trees at a height of 20–40 cm above the ground in areas favored for mammal detection and as close as possible to predefined grid locations (*Cusack et al., 2015*). Surveys were conducted in the *terra firme* forest during the dry season, which was defined as the months with <100 mm average rainfall (*Espinoza Villar et al., 2009*), and in the *igapó* forest during the low-water phase. The cameras operated 24 h per day and were programmed to successively take three photographs and record 10-s videos, with no delay between subsequent triggers. No lures or baits were used as the aim was to capture the natural behavior of the species (*Burton et al., 2015*).

Camera traps are consistently able to detect terrestrial mammals ≥0.5 kg (*Rovero, Tobler & Sanderson, 2010*). Data management and estimates of the number of independent records were performed using the *camtrapR* package (*Niedballa et al., 2016*). Species nomenclature followed the guidelines of *Quintela, Da Rosa & Feijó (2020)*. Congener armadillos (*Dasypus* spp.) were treated as single species because of the difficulty in differentiating them on nocturnal (black and white) images or videos. The species were grouped in six functional groups (frugivores/herbivores, frugivores/granivores, frugivores/omnivores, insectivores/omnivores, insectivores, and carnivores), which were adapted from *Paglia et al. (2012)*. *Myrmecophaga tridactyla*, *Tamandua tetradactyla*, and *Priodontes maximus* were reassigned from myrmecophages to insectivores, *Hydrochoerus hydrochaeris* from herbivores tofrugivores/herbivores, *Pteronura brasiliensis* from piscivores to carnivores, and *Eira barbara* from carnivores/omnivores to carnivores to avoid trophic categories with only one species.

## Data analysis

### Attributes and composition of the mammal assemblages

The attributes of the assemblages were compared first, *i.e.*, number of species observed, evenness, abundance and biomass, per camera trap, between the *igapó* and *terra firme* forests using the non-parametric Wilcoxon signed-rank test because the data distribution was not normal (*Woolson, 2007*). Evenness values were calculated using the Pielou index (J'), which was derived from the Shannon index, using the *vegan* package (*He & Legendre, 2002*; *Oksanen et al., 2014*). This evenness measure was selected because it is an excellent predictor of species abundance and richness in tropical forests (*Stocker, Unwin & West, 1985*). J' values ranged from 0.0 to 1.0, with higher values representing greater distribution of species. For the abundance analysis, the camera-trapping rate (number of sampling points for independent detections to 100 trap days) was used (*Rovero & Marshall, 2009*). The sampling effort was assessed by the number of days for which each camera was operational throughout the session. The abundance data were used both for species-level

comparisons between forest types and for subsequent compositional analyses. Photographs or videos of the same species at the same point taken >60 min apart were defined as independent detection events (*Meek et al., 2014*). The relative biomass was calculated by multiplying the camera-trapping rate by the mean body mass and mean group size for gregarious species (*Rosa et al., 2021*). Mean body mass data were obtained from the EltonTraits1.0 database (*Wilman et al., 2014*).

To graphically evaluate rank abundance curves, the relative abundance index (RAI = overall number of independent detections per 1,000 trap days) was used. For this analysis, the overall abundance of each species per forest type was considered (*O'Brien, 2011*), and the sampling effort was the sum of the days during which each camera was operational throughout the campaign in each forest type.

Subsequently, the rarefied species richness was estimated in both forest types (hereafter referred to as "species richness"), using rarefaction curves with the first Hill number (species richness, q = 0) (*Chao et al., 2014*). Rarefaction was estimated as the mean of 1,000 replicate bootstrapping runs to estimate 95% confidence intervals (*Colwell, Mao & Chang, 2004*). This approach accounts for differences in the sampling effort between camera-trap locations (*i.e.*, variation in deployment duration due to units being set and collected at different times) and does not require data to be discarded. The sample-based interpolated and extrapolated values were generated in the *iNEXT* package (*Hsieh, Ma & Chao, 2016*). The accumulation curves were computed based on the time taken to accumulate new species and reach an asymptote. The species richness between forest types differed significantly ($p < 0.05$) when the 95% confidence intervals did not overlap (*Colwell et al., 2012*), and when the curves overlap, 84% confidence intervals instead of 95% can be used to represent any statistical significance at the level of 0.05, as recommended by *MacGregor-Fors & Payton (2013)*. In addition, the sampling effort was assessed as to whether it was sufficient for determining mammal species diversity using these curves.

To visualize the composition of the mammalian assemblages of the *terra firme* and *igapó* forests, the metaMDS function of the *vegan* package was used to reduce the matrix dimensionality of the mammalian species recorded using non-metric multidimensional scaling (NMDS) ordination with two axes (*Anderson, 2014*; *Legendre & Legendre, 1998*). Before the analyses, the camera trap station weights were standardized by dividing the number of recordings of each species (camera trapping rate) in each matrix cell by the total number of recordings performed by the camera-trap station (decostand function, MARGIN = 1) in order to reduce the discrepancy between sites with abundant species.

Subsequently, the differences in assemblage structure between the forest types (centroids of the groups) were statistically tested using permutational multivariate analysis of variance (PERMANOVA, adonis function, *vegan* package) (*Anderson, 2014*). The $R^2$ value indicated the effect of each forest type on the species composition. In this study, higher $R^2$ values indicated greater differences in species composition between the forest types. PERMANOVA and NMDS were performed using a Bray–Curtis dissimilarity index.

To detect the spatial autocorrelation in the trapping records between sampling points, spatial coordinates of trapping points computed as a Euclidean matrix were compared with the species abundance computed as a Bray–Curtis dissimilarity matrix for each forest type

(*Legendre & Legendre, 1998*). The correlation between both matrices was analyzed *via* a Mantel test with 9,999 permutations. Moreover, to evaluate differences in within-group variation (dispersion), an analysis of homogeneity of multivariate dispersions (PERMDISP, betadisper function, *vegan* package) was used (*Anderson & Walsh, 2013*). This test is based on the Euclidean distances between sampling units and their centroids in a two-dimensional PCoA diagram. *p*-values were obtained using F-tests based on sequential sums of squares from 10,000 permutations. Additionally, the beta diversity was calculated and partitioned into components of balanced variation in abundance and abundance gradients and into spatial turnover and nestedness-resultant for each forest type. Using the abundance-based (Bray-Curtis dissimilarity) index, the beta diversity was quantified for multiple site dissimilarly ($\beta_{BC}$). In this approach, the index is partitioned in two components: the balanced variation ($\beta_{BAL}$) and abundance gradient ($\beta_{GRA}$). $\beta_{BAL}$ evaluates the dissimilarity of pair of samples, taking into account the contribution of similar abundances in different species. $\beta_{GRA}$ quantifies between a pair of samples for the same species the contribution of differences in their abundance, and reflects the increase or reduction between communities (*Baselga, 2017*). For the incidence-based (Sorensen dissimilarity) index ($\beta_{SOR}$) the original abundance-based community data was transformed into a presence/absence matrix in order to calculate the nestedness-resultant ($\beta_{NES}$) and the turnover ($\beta_{SIM}$) components, with the latter representing the number of species that are replaced between sites in relation to the total number of species that could be replaced (*Baselga, 2010*). The index employed the *betapart 1.3* package using the function beta.sample.abund and beta.sample, respectively (*Baselga et al., 2018*).

Finally, the envfit function (*vegan* package) with 999 permutations was applied to the species detections and ordination axis scores in order to identify the key species that contributed the most to the variation in assemblage structure between camera trap locations (*Oksanen et al., 2014*). $R^2$ indicated the contribution of each species to the variation in the assemblage structure between camera-trap locations. All analyses were conducted in R software version 4.1.2 (*R Development Core Team, 2016*).

### Estimates of occupancy (Ψ) and detection (p) of species

Although the relative abundance of data allows comparisons, the occupancy and detection probability were also estimated for each species and functional group. The camera trap data were used to assess the differences in the occurrence of mammals in relation to forest types using an approach based on the occupancy model (*Mackenzie et al., 2002*). Such models consider imperfect detection and used repeated presence–absence surveys (detection histories) at multiple sampling units in order to estimate the detection probability (*p*) and the true proportion of area occupied by a species ($\psi$) (*Mackenzie et al., 2017*).

To model species and functional group occurrence, single species occupancy models implemented in a Bayesian framework were used (*Welsh, Lindenmayer & Donnelly, 2013*). A set of models was tested for both species and functional groups. Models were run using the R package *ubms* version 1.0.2 (*Kellner, 2021*). Due to the short interval between samplings, and because there was no interest in the transition probabilities, it was assumed

that occupancy of sites was constant between the sampling years within each forest type. Then, the "stacked" approach was applied (*Kellner et al., 2022*). Only species observed in more than 25% of the camera-trap stations were included in the occupancy analyses (twenty species). Detection histories of mammals were collapsed into multiple-day sampling on eight occasions to maximize the detection estimates and improve maximum likelihood convergence. In our study, the occupancy likely represented the probability of site use rather than being the true occupancy as the independence assumption was not met since the distance between sampling units was one kilometer, which is considered a short distance for some species like jaguars (*Burton et al., 2015*; *Kays et al., 2020*). Therefore, "habitat use" is used instead of "occupancy" throughout this paper (*Mackenzie et al., 2017*).

Six model combinations for each species were fitted with the categorical variable forest type [for.typ], which influenced the occupancy and detection [model $\psi$(for.typ) $p$(for.typ)] and the variable effort [eff] and affected only the detection [model $\psi$(.) $p$(eff)]. The variable effort referred to the number of days that each camera-trap station was active within occasions. This was performed to account for the possible failure of camera-trap units and improve the detections. In order to allow comparisons, the same model structure was maintained between the species (*Mackenzie et al., 2017*).

The procedures below followed those recommended by *Nguyen et al. (2022)*, with the default vague priors used for all models (normal distribution with mean = 0 and SD = 10). Four parallel Markov chains with 5,000 iterations each were run, and the first 2,500 iterations were discarded as burn-in. The model convergence was assessed by checking that Rhat equaled 1 and by visually examining the model traceplots (*Gelman et al., 2014*). To determine the significance of the covariate effects, 95% and 75% credible intervals of the posterior distribution were generated. A coefficient was considered to have strong support if its 95% Bayesian credible interval did not overlap zero (*Nguyen et al., 2022*).

The different candidate occupancy models obtained for each species were compared and ranked using expected log pointwise predictive density (elpd) as a measure of predictive accuracy of the models. To assess the model predictability compared with that of the top model, the pairwise differences in elpd ($\Delta$elpd) between each model and the top model and the standard errors thereof (SE [$\Delta$elpd]) were calculated (*Nguyen et al., 2022*). A model was considered to be less accurate than the top model if its absolute difference in elpd was greater than the standard error of that difference (*Doll & Jacquemin, 2019*; *Kellner, 2021*).

## RESULTS

### Species richness and relative abundance

A total sampling effort of 10,743 camera trap days in four campaigns was obtained: two for the *terra firme* forest (dry season: 6,013 trap days) and two for the *igapó* forest (low-water phase: 4,730 trap days), and included 31 non-volant mammals from eight taxonomic orders. Among the recorded species, five are classified as Vulnerable and two as Near threatened (NT) in the IUCN Red List of Threatened Species (*IUCN, 2022*). The remaining species are classified as Least concern (LC) (Data S1). Four times more non-volant mammal recordings per total effort were obtained in the *terra firme* forest (0.97

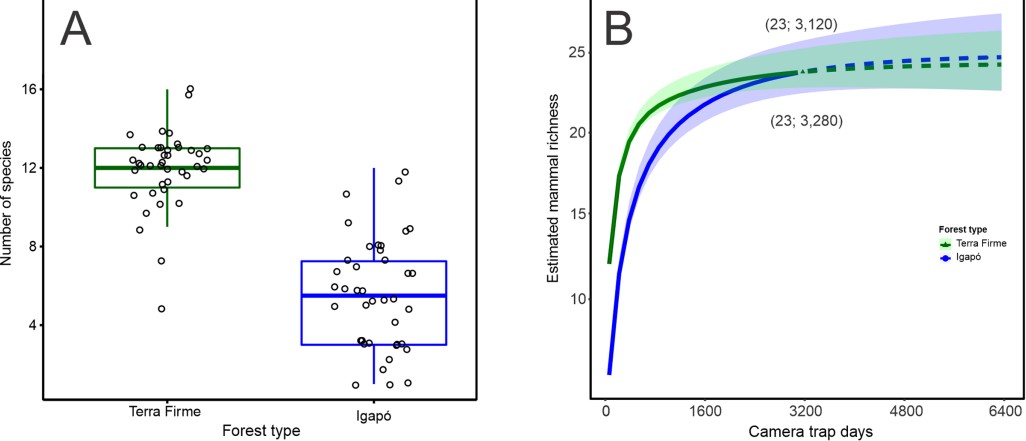

**Figure 2 Observed and estimated richness of terrestrial mammal species in the Cuieiras River basin, Central Amazon, Brazil.** (A) Difference in the observed number of species per camera trap point between *terra firme* and *igapó* forests (Wilcoxon: W = 1,534, *p* < 0.001). (B) Species accumulation curves (interpolated and extrapolated) of terrestrial mammal assemblage in both forest types. The solid lines represent the cumulative number of species as a function of the number of camera trap points sampled in *terra firme* and *igapó* forests. Dashed lines indicate extrapolated species numbers with a standardized number of camera trap points (40) in each forest type and the shaded areas represent 84% confidence intervals with overlapping indicating that there was no statistically difference in richness between both forest types.

independent records/traps*day) than in *igapó* forest (0.27 independent records/traps*day). Five species including two primate (*Saimiri sciureus* and *Sapajus apella*), two marsupial (*Marmosa* spp. and *Oecomys* spp.), and one rodent (*Guerlinguetus aestuans*) species were considered as eventual species (primates detected by camera traps are predominantly arboreal and these marsupial and rodents are small and not efficiently detected by this sampling method) and were not included in the analysis.

The observed number of species per camera trap point ranged from 5 to 16 species in the *terra firme* forest. It was greater than that observed in the *igapó* forest, where 1 to 12 species per trap point were observed (Wilcoxon: W = 1,534; *p* < 0.001; Fig. 2A). The camera-trapping rate (number of records per camera effort) in the *terra firme* forest was on average 100.10 (range = 30.07–283.33), whereas it was 27.56 (range = 0.72–113.63) in the *igapó* forest (Wilcoxon: W = 1,522; *p* < 0.001; Fig. S1A). The total relative biomass in *terra firme* (30,469; range = 160.66–2,813.33) was twice as much as that in the *igapó* (15,711; range = 0.96–2,900.29) forest (Wilcoxon: W = 1,226; *p* < 0.005; Fig. S2B). There were differences in estimates of evenness between the two forest types (Wilcoxon: W = 337; *p* < 0.006), which suggests dissimilar relative abundance distributions of rare and common species per camera-trap point, with an average greater value of evenness in the *igapó* forest (0.70; range = 0.41–1.00) compared to the *terra firme* (0.50; range = 0.33–0.76) forest (Fig. S1C). However, when pooled data were considered, the overall evenness was similar between both forest types (*terra firme* = 0.34; SE ± 0.01 and *igapó* = 0.35; SE ± 0.03).

When the species richness estimate was standardized to the sample size, overall, the species richness was similar in the *igapó* (24 species SE ± 1.83) and *terra firme* (23 species SE ± 1.29) forests (Fig. 2B). Additionally, the same completeness of 99% was found for

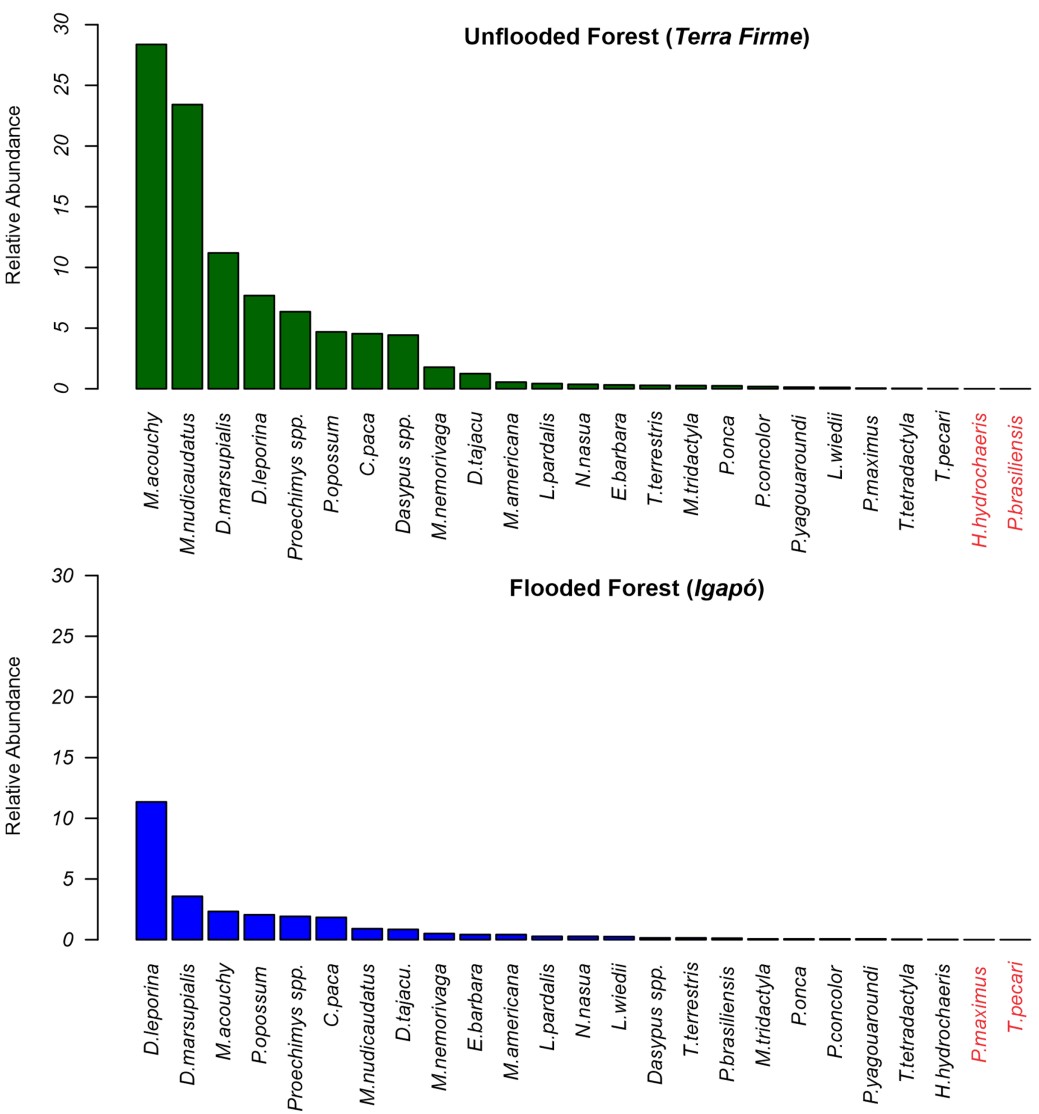

**Figure 3 Relative abundance rates of terrestrial mammals recorded in *terra firme* and *igapó* forests.** Scientific names in red markers mean absence records during the sampling of the species in the forest type.

both forest types. Rarefaction curves reach an asymptote for both environments, with a similar effort of 3,120 and 3,280 camera*trap days for the *terra firme* and *igapó* forests, respectively. This outcome indicates that the sampling effort was enough to represent all the species of the mammal fauna in the study sites.

At the *terra firme* sites, the two most frequently recorded mammals accounted for 53.6% of all recordings, with the frugivore/granivore *Myoprocta acouchy* (RAI = 28.65) being the most common species, and the insectivore/omnivore *Metachirus nudicaudatus* (RAI = 23.43) being the second most common. In contrast, in the *igapó* forest, *Dasyprocta leporina* was the dominant species (RAI = 11.35; a proportion of 41% of all captures) and *Didelphis marsupialis* (RAI = 3.57) the second most dominant (Fig. 3).

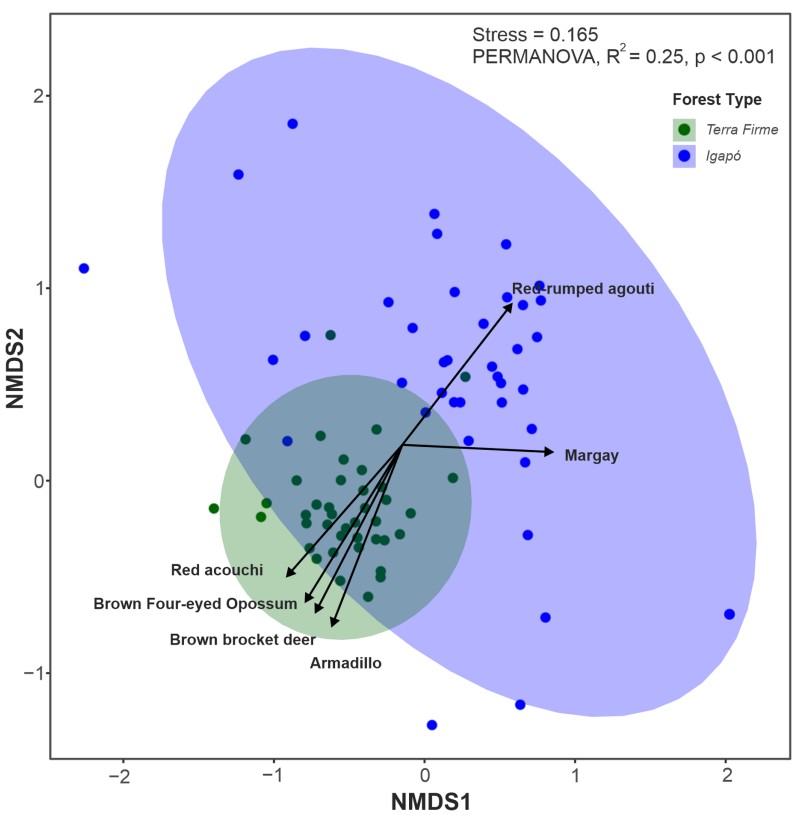

**Figure 4 Non-metric multidimensional scaling (NMDS) ordination plot of the terrestrial mammal species composition detected by camera traps in Amazonian *igapó* and *terra firme* forests.** NMDS plots area based on the dissimilarity matrix derived from relative abundance data for each species. The species represented in the graph contributed the most to the variations in assemblage structure between camera trap locations in each forest type.

## Structure of mammal assemblages

Among the 26 species evaluated, 22 (84%) were found in both forest types, two (*Tayassu pecari* and *Priodontes maximus*) were recorded only in the *terra firme* forest and two others (*Hydrochoerus hydrochaeris* and *Pteronura brasiliensis*) were seen only in the *igapó* forest. Although most species occupied both forest types, the multivariate analyses along the first two axes (85% of variation explained) revealed that *terra firme* and *igapó* forests formed distinct clusters in the NMDS ordination (Fig. 4). This was further confirmed by permutation tests (PERMANOVA: F = 25.45; $p = 0.001$).

The Mantel test between geographic distances and species Bray–Curtis dissimilarity between forest types was significant for the *terra firme* forest: Mantel test $r = 0.16$; $p = 0.01$, though not for the *igapó* forests: (Mantel test $r = 0.08$; $p = 0.09$), thus indicating that no spatial autocorrelation occurred in the trapping records between camera-trapping points in the latter. When within-group variation was evaluated, the *igapó* forest occupied the largest area in community space, whereas camera-trap locations in the *terra firme* forest were relatively similar to each other (PERMDISP: F = 35.23, $p < 0.001$). Variation in the species composition assemblages, or beta diversity, was dissimilar for both forest types ($\beta_{BRA} = 0.75$; SD ± 0.02 in the *terra firme* forest and $\beta_{BRA} = 0.85$; SD ± 0.02 in the *igapó*

forest) when the abundance based-dissimilarity was considered. Most of this beta diversity derived from the balanced variation component ($\beta_{BAL}$ = 0.58; SD ± 0.03 in the *terra firme* forest and $\beta_{BAL}$ = 0.63; SD ± 0.07 on the *igapó* forest). The abundance-gradient component was very low for both environments ($\beta_{GRA}$ = 0.16; SD ± 0.03 in the *terra firme* forest and $\beta_{GRA}$ = 0.21; SD ± 0.06 in the *igapó* forest). As well as with the incidence data with a higher turnover ($\beta_{SOR}$ = 0.80; SD ± 0.02) in the *igapó* compared to the *terra firme* ($\beta_{SOR}$ = 0.59; SD ± 0.03), with these differences derived from the turnover component ($\beta_{SIM}$ = 0.50; SD ± 0.03 in the *terra firme* forest and $\beta_{SIM}$ = 0.67; SD ± 0.05 in the *igapó* forest).

The key species with assemblage differences between both forest types identified by envfit analysis were *Metachirus nudicaudatus* ($R^2$ = 0.37; $p < 0.001$), *Myoprocta acouchy* ($R^2$ = 0.31; $p < 0.001$), *Dasypus* spp. ($R^2$ = 0.28; $p < 0.001$), and *Mazama nemorivaga* ($R^2$ = 0.13; $p < 0.01$) in the *terra firme* forest and *Dasyprocta leporina* ($R^2$ = 0.13; $p < 0.002$) and *Leopardus wiedii* ($R^2$ = 0.12; $p < 0.006$) in the *igapó* forest (Data S2).

## Estimate of the occupancy ($\Psi$) and probability of detection ($p$)

All the models that analyzed the functional groups revealed that the variable "forest type" influenced the occupancy parameter (Table 2). This did not occur when the occupancy for species was evaluated separately (Data S3). The variable "effort" was an important detector predictor for six species, with an increase in the effort improving the probability of detection. The occupancy varied among species from $\psi$ = 0.30; SD ± 0.18 for the small carnivore *Puma yagouaroundi* to $\psi$ = 0.97; SD ± 0.02 for the frugivore/granivore *Myoprocta acouchy* in the *terra firme* forest. In the *igapó* forest, the lowest occupancy was $\psi$ = 0.16; SD ± 0.07 for *Nasua nasua* and the highest was $\psi$ = 0.79; SD ± 0.05 for *Dasyprocta leporina*. The occupancy was estimated for twenty species and similar occupancies between the *terra firme* and *igapó* forests (Fig. S2) were found for eight species (40%), which were mostly represented by the carnivore guild. In contrast, the greatest variations were found for the insectivores/omnivores, with a higher occupancy in the *terra firme* forest even when *igapó* environments were available (Fig. S3). Regarding the occupancy of species, the greatest difference between forest types was found for the frugivore/granivore *Myoprocta acouchy*, which avoided the *igapó* areas, whereas the carnivore *Leopardus wiedii* similarly occupied both forest types and was the second species with the highest occupancy in the *igapó* forest ($\psi$ = 0.74; SD ± 0.17).

## DISCUSSION

Our study provides evidence that terrestrial mammal assemblages are structured in adjacent, yet very distinct, unflooded (*terra firme*) and seasonally flooded (*igapó*) forests in the Central Amazon. Most *igapó* areas are rather small (*igapós* are seldom more than 100–300 m in width) and usually adjacent to *terra firme* areas along forest streams. This organization allows the lateral movement of terrestrial species during low-water phases (*Hawes & Peres, 2014*; *Wittmann & Junk, 2016*). The phenology of these forest types evidences two asynchronous peaks of leaf flushing and fruit production (*Parolin, Waldhoff & Piedade, 2010*). Thus, seasonal movements of mammal species were expected to occur between those peaks as individuals track resources. Lateral movements have been reported
**Table 2 Model selection for the functional groups with all combinations of covariates of occupancy models.** Models were ranked with expected predictive accuracy (elpd) values calculated using a leave-one-out cross-validation approach. Δelpd represents the pairwise differences in elpd (relative to the top model). ΔSE is the standard error of Δelpd. K is the effective number of parameters and weight is the weight of each variable influencing the parameter. Occupancy ($\psi$) and probability of detection (p) between terra firme and igapó forests are indicated.

| Functional Group | Model | elpd | K | Δelpd | ΔSE | Weight | $\psi$ Terra firme | Igapó | p Terra firme | Igapó |
|---|---|---|---|---|---|---|---|---|---|---|
| Carnivores | $\psi$ (for.typ) p (for.typ) | −359.87 | 1.57 | 0.00 | 0.00 | 0.81 | 0.97 ± 0.03 | 0.94 ± 0.05 | 0.08 ± 0.01 | 0.05 ± 0.01 |
| | $\psi$ (.) p (for.typ) | −359.89 | 1.59 | −0.01 | 0.15 | 0.00 | | | | |
| | $\psi$ (for.typ) p (eff) | −360.68 | 1.57 | −0.81 | 1.62 | 0.17 | | | | |
| | $\psi$ (.) p (eff) | −360.88 | 1.54 | −1.01 | 1.76 | 0.01 | | | | |
| | $\psi$ (for.typ) p (.) | −361.40 | 1.08 | −1.53 | 1.54 | 0.00 | | | | |
| | $\psi$ (.) p (.) | −361.43 | 0.92 | −1.56 | 1.70 | 0.00 | | | | |
| Insectivores | $\psi$ (for.typ) p (.) | −126.19 | 2.32 | 0.00 | 0.00 | 0.62 | 0.75 ± 0.15 | 0.28 ± 0.13 | | |
| | $\psi$ (for.typ) p (eff) | −126.26 | 2.21 | −0.07 | 0.07 | 0.37 | | | | |
| | $\psi$ (for.typ) p (for.typ) | −127.40 | 3.74 | −1.21 | 0.92 | 0.00 | | | 0.03 ± 0.01 | 0.04 ± 0.03 |
| | $\psi$ (.) p (for.typ) | −129.04 | 3.46 | −2.85 | 2.17 | 0.00 | | | | |
| | $\psi$ (.) p (eff) | −130.11 | 1.94 | −3.92 | 2.13 | 0.00 | | | | |
| | $\psi$ (.) p (.) | −130.51 | 1.90 | −4.31 | 2.14 | 0.00 | | | | |
| Insectivores/Omnivores | $\psi$ (for.typ) p (for.typ) | −689.44 | 8.81 | 0.00 | 0.00 | 0.87 | 0.97 ± 0.02 | 0.66 ± 0.08 | 0.71 ± 0.01 | 0.17 ± 0.02 |
| | $\psi$ (.) p (for.typ) | −699.03 | 9.64 | −9.59 | 3.07 | 0.00 | | | | |
| | $\psi$ (for.typ) p (eff) | −830.43 | 9.35 | −140.99 | 33.00 | 0.10 | | | | |
| | $\psi$ (.) p (eff) | −842.55 | 9.31 | −153.11 | 31.79 | 0.00 | | | | |
| | $\psi$ (for.typ) p (.) | −851.12 | 8.87 | −161.68 | 33.54 | 0.02 | | | | |
| | $\psi$ (.) p (.) | −863.49 | 9.29 | −174.05 | 32.36 | 0.00 | | | | |
| Frugivores/Granivores | $\psi$ (for.typ) p (for.typ) | −753.59 | 8.45 | 0.00 | 0.00 | 0.65 | 0.97 ± 0.01 | 0.85 ± 0.05 | 0.82 ± 0.01 | 0.55 ± 0.02 |
| | $\psi$ (.) p (for.typ) | −757.30 | 8.56 | −3.71 | 1.43 | 0.00 | | | | |
| | $\psi$ (for.typ) p (eff) | −791.32 | 7.01 | −37.72 | 21.50 | 0.34 | | | | |
| | $\psi$ (.) p (eff) | −794.82 | 6.71 | −41.22 | 21.42 | 0.00 | | | | |
| | $\psi$ (for.typ) p (.) | −810.86 | 5.99 | −57.26 | 21.74 | 0.00 | | | | |
| | $\psi$ (.) p (.) | −814.51 | 5.96 | −60.91 | 21.66 | 0.00 | | | | |
| Frugivores/Herbivores | $\psi$ (for.typ) p (for.typ) | −807.03 | 6.69 | 0.00 | 0.00 | 0.56 | 0.97 ± 0.02 | 0.74 ± 0.06 | 0.35 ± 0.01 | 0.24 ± 0.02 |
| | $\psi$ (for.typ) p (eff) | −808.66 | 5.22 | −1.63 | 6.82 | 0.43 | | | | |
| | $\psi$ (for.typ) p (.) | −812.64 | 3.84 | −5.61 | 6.82 | 0.00 | | | | |
| | $\psi$ (.) p (for.typ) | −813.89 | 7.05 | −6.86 | 2.10 | 0.00 | | | | |
| | $\psi$ (.) p (eff) | −815.62 | 4.98 | −8.59 | 7.11 | 0.00 | | | | |
| | $\psi$ (.) p (.) | −820.00 | 4.12 | −12.97 | 7.04 | 0.00 | | | | |
| Frugivores/Omnivores | $\psi$ (for.typ) p (for.typ) | −809.30 | 8.70 | 0.00 | 0.00 | 0.51 | 0.95 ± 0.03 | 0.67 ± 0.07 | 0.45 ± 0.01 | 0.26 ± 0.02 |
| | $\psi$ (.) p (for.typ) | −815.87 | 7.70 | −6.57 | 3.17 | 0.05 | | | | |
| | $\psi$ (for.typ) p (eff) | −816.93 | 6.48 | −7.62 | 11.40 | 0.43 | | | | |
| | $\psi$ (.) p (eff) | −823.90 | 5.65 | −14.59 | 11.61 | 0.00 | | | | |
| | $\psi$ (for.typ) p (.) | −826.64 | 5.74 | −17.33 | 11.77 | 0.00 | | | | |
| | $\psi$ (.) p (.) | −833.62 | 4.91 | −24.32 | 11.96 | 0.00 | | | | |

for different animals such as primates (*Haugaasen & Peres, 2005c*), bats (*Bobrowiec et al., 2014*), small rodents (*Pereira et al., 2013*), and other groups of terrestrial vertebrates (*Costa, Peres & Abrahams, 2018*; *Alvarenga et al., 2018*). However, these studies compared different environments, *i.e.*, *terra firme* and nutrient-rich flooded *várzea* forests. To our knowledge, the present study is the first to investigate terrestrial mammals in unflooded *terra firme* and nutrient-poor flooded blackwater *igapó* forests using camera trapping. Our data provide evidence that terrestrial mammal assemblages are structured differently in these forest types. These findings emphasize the relevance of environmental heterogeneity mediated by different vegetational formations in the Amazon.

## Species diversity in both forest types

As expected for the patterns of diversity in nutrient-poor habitats (*Zapata-Ríos et al., 2021*), a marked difference was found in the abundance of mammals, biomass, and evenness between forest types, whereas the richness was similar. The relative abundance per camera-trap point and the relative biomass were greater in the *terra firme* than those in the *igapó* forest. To explain such patterns, we considered two possible explanations. First, the influence of the nutrient load in flood water on the mammal assemblages seemed to be particularly important for the total biomass, which was twice as abundant in *terra firme* than in the *igapó* forest. Soil fertility is a key driver of forest dynamics in terms of productivity, tree turnover, and cation exchange capacity, and is strongly associated with the presence of arboreal and terrestrial mammals in the Amazon (*Buendía et al., 2018*; *Ferreira Neto et al., 2021*). This association involves bottom-up mechanisms by which the limitation of soil nutrients affects the cost-effectiveness of plant investments in reproduction and fruiting bodies (*Chave et al., 2010*). This is linked to lower productivity, which results in lower carrying capacity (*Hawes et al., 2012*; *Junk et al., 2015*). The other explanation for the observed patterns can be related to a greater dispersion of individuals, given that animals are now spread over a greater area what could have caused this dilution effect that was reflected in a lower abundance (*Haugaasen & Peres, 2007*; *Hawes & Peres, 2014*).

In contrast, the equitability per camera trap measured by the Pileou index was greater in the *igapó* forest. This index assesses the diversity, which combines species richness and equitability (*i.e.*, the evenness of the distribution of individuals among a species). This was in line with the observation that naturally nutrient-depleted habitats tend to harbor assemblages with more evenness among the species present (*Tilman, Kilham & Kilham, 1982*). However, the species frequencies also tend to overestimate the abundances of rare species and underestimate the abundances of widespread species, leading in some cases to a relatively low variation in the resulting species evenness measure (*Magurran, 2004*), as observed for the overall evenness between both forest types. Therefore, as recommended by other authors the assessment of species evenness should be used more–as a complement to the assessment of species richness–in monitoring projects and empirical studies (*Reitalu et al., 2009*).

Local species richness is also limited by the available species pool, *i.e.*, richness in a given community or site is determined by the number of species available at the next largest scale

(*Zobel, 1997*). Although the observed richness varied between our sample points, this apparent difference in the number of species reflected the natural heterogeneity of each point and larger effort in *terra firme* forest. When the data were extrapolated to the same effort/coverage, the overall richness of assemblages was similar in both forest types since, in our study area, both environments are supplied by the same regional species pool, although the *terra firme* forest has a more stratified forest structure (*Wittmann, Schöngart & Junk, 2010*; *Zapata-Ríos et al., 2021*). Nevertheless, the total number of species recorded in the Cuieiras River basin were unexpectedly similar to those described in other camera-trap-based studies carried out in the richer productivity habitats of *várzea* and *terra firme* forests in the Amazon (*Haugaasen & Peres, 2005a*; *Tobler et al., 2008*; *Alvarenga et al., 2018*; *Costa, Peres & Abrahams, 2018*; *Rosa et al., 2021*). This further indicated that the sampling effort was sufficient.

The results should be treated with caution due to the limitation in our sampling design. During our sampling, it was not possible to concurrently install camera traps in *terra firme* areas away from *igapó* areas and compare them with the *terra firme* areas adjacent to the *igapó* areas. However, it is important to acknowledge the severe logistical limitations to carrying out this kind of study, which is based on deploying several camera traps by walking through large areas within a rough landscape of remote and dense forests. Nevertheless, it was possible to obtain relevant statistical outputs that allowed for a first exploration of the proposed hypotheses.

## Spatial patterns of assemblage structure

Our results appear to be in agreement with the lateral movement hypothesis, which states that terrestrial mammal species move from *terra firme* forests to exploit seasonally available food resources in *igapó* forests during the low-water phase. The results also revealed a marked difference in mammal assemblages between the forest types.

As previously reported, some studies comparing flooded and unflooded forest types have shown that food availability, more specifically fruit production and distribution within habitats, was the most important variable in influencing mammal occupancy and abundance (*Pereira et al., 2009*; *Hawes & Peres, 2014*; *Antunes et al., 2019*). Our data revealed that in terms of abundance the frugivores/granivores *Myoprocta acouchy* and *Dasyprocta leporina* were primarily responsible for the differences in assemblage composition contrary to our expectations. In fact, the red acouchi is associated with primary forests and 90% of its diet consists of fleshy fruits (*Dubost, 1988*). In contrast, the dominant species in the *igapó* forests, the red-rumped agouti, despite being from the same functional group, is more generalist and is able to adapt to different types of environments thanks to the flexibility in its diet (*Dubost & Henry, 2006*; *Jorge, 2008*). The occupancy in the *igapó* forest was low for other functional guilds, such as the insectivores/omnivores represented by *Dasypus* spp. and *Metachirus nudicaudatus*, probably because these species strongly rely on ground and understory vegetation for feeding or sheltering (*Pereira et al., 2013*). Other guilds, such as the carnivores, thrived well in both forest types probably because they have large home ranges and take advantage of their prey's movements from the *terra firme* to *igapó* forests (*Mendes Pontes & Chivers, 2007*; *Antunes et al., 2019*).

For example, the occupancy of both forest types by the small felid *Leopardus wiedii*, usually found in more closed environments (*Tortato et al., 2013*), was already somewhat expected, since this feline might alter its use of space depending on prey availability and also considering its great arboreal adaptations (*Alvarenga et al., 2018*). However, future studies should include camera-trap sampling in the canopies of *igapó* forests during the high-water level, since it is known that some felids, like jaguars, have a unique lifestyle in *várzea* forests and use the trees for extended periods of the year (*Ramalho et al., 2021*), as well as some scansorial species such as opossums, tayras, coatis and tamanduas are needed to confirm the dependency of these vertebrate species when *igapós* forest are inundated by the flood waters.

The distribution of species in each forest type was also different within the groups, with higher dispersion between camera trap points in the *igapó* forests. In this forest type, as the sample points became physically more separated, their corresponding mammal communities become more dissimilar. In parallel, our results showed that the high beta diversity was explained by both the balanced variation in abundance ($\beta_{BAL}$) and the turnover ($\beta_{SIM}$) components. This means that the compositional dissimilarity in the *igapós* derived both from changes in species abundance and turnover from site to site, with different sets for different sites. In other words, the individuals of a given species at one site were substituted by an equivalent number of individuals of a different species at another site (*Baselga, 2010*; *Baselga, 2017*).

This discrepancy or turnover between sampling sites is likely related to heterogeneity in vegetation structure even within the same forest type (*Tews et al., 2004*; *Junk & Piedade, 2010*). In the Central Amazon, the topography of the *terra firme* forest is considered a strong predictor of vegetation structure at a local scale (*Costa et al., 2009*). In *igapó* forests, the vegetation structure is strongly linked to the duration of the annual inundations (*Wittmann, Schöngart & Junk, 2010*), in which some areas of the forest may be dry for less than four weeks a year, whereas other areas remain unflooded for 10–12 weeks (*Junk et al., 2011*). This phenomenon was observed during our sampling that lasted for about three months, since some cameras were found submerged at the time of their removal. Thus, our findings were consistent with the idea that terrestrial mammals temporarily occupy *igapó* sites during the low-water phase as many seeds and fruits that had been floating in the water are then exposed on the forest floor. In contrast, rising floodwaters force several species to seek suitable habitats in more stable upland forests (*Haugaasen & Peres, 2005b*; *Parolin, Wittmann & Ferreira, 2013*).

## Conservation implications

Terrestrial mammals form a key group for forest functionality and provide vital services in flooded and non-flooded ecosystems in the Amazon (*Haugaasen & Peres, 2005a*), mainly in nutrient-poor soil habitats where these animals may significantly contribute to dissipating the large-scale P gradient from rivers and seasonally flooded and *terra firme* forests (*Buendía et al., 2018*; *Ferreira Neto et al., 2021*). Improving our understanding of the spatial distribution of mammal assemblages might help us to predict the persistence of species and safeguard them in low resilience ecosystems such as those in flooded areas

(*Castello et al., 2013*; *Flores et al., 2017*). Additionally, the low abundance of species in the *igapó* forest does not imply that this habitat should be neglected during conservation planning. On the contrary, various studies have shown that both aquatic and terrestrial species use seasonally flooded blackwater forests, even though they represent a small portion (~2%) of the Amazonian environment (*Saint-Paul et al., 2000*; *Montero, Piedade & Wittmann, 2014*; *Laranjeiras, Naka & Cohn-Haft, 2019*; *Pringle et al., 2019*). This suggests that the complementary areas of habitats are crucial to the long-term maintenance of viable populations. Nevertheless, despite the importance of these environments, the management policies in Brazil principally protect terrestrial ecosystems, whereas floodplain forests remain poorly represented (less than 1% are strictly protected) in the protected areas of the Amazon basin (*Latrubesse et al., 2021*). Therefore, it is necessary to reinforce the importance of maintaining the mosaic of natural habitats in the entire landscape, with the major drainage basins representing management units. This agrees with what is proposed for the mosaic of Protected areas of Negro River (*Didier et al., 2017*). These mosaics are essential for maintaining a range of ecological processes, *e.g.*, nutrient transport, lateral movements of species, and persistence of apex predators.

## CONCLUSIONS

Similar species richness was observed in both the *igapó* and *terra firme* forests; however, higher species abundance and biomass were found in the *terra firme*, and evenness was greater in the *igapós*. A marked difference was noted in the composition of terrestrial mammal species when comparing both forest types, which was then explored in more detail for 20 mammal species in terms of habitat use. The observed differences were associated with the higher abundance of some specific functional groups, *i.e.*, frugivores/granivores, rather than with the species turnover. During the low-water phase, many seeds and fruits that had been floating in the water in the high-water phase are left exposed in the forest, thus allowing their consumption by terrestrial species. As a result, mainly frugivores annually move into the *igapó* forests to take advantage of these pulses of food resource availability.

These findings additionally indicate that seasonal flooding is an important factor in structuring the composition of mammal assemblages in *terra firme* and floodplain forests, even in nutrient-poor habitats. Our study provides a better understanding of the mammal fauna, its ecological specificities and assemblage structure across different forest types in the Central Amazon and, together with the characterization in future studies of other ecological drivers and sampling designs not evaluated here, this will help to better elucidate the explicit patterns of the faunal structure in these seasonal habitats.

## ACKNOWLEDGEMENTS

The authors are grateful to the Museu na Floresta, an agreement of Instituto Nacional de Pesquisas da Amazonia (INPA), Japan Science and Technology Agency, Japan International Cooperation Agency, and Kyoto University for the logistical support. The authors would like to thank the field assistants who helped with the camera-trapping

campaigns. The dataset was provided by Grupo de Pesquisa em Mamíferos Amazônicos (GPMA/INPA). This is publication No. 61 of the Amazonian Mammals Research Group.

### Funding

This work was financed by the Coordenação de Aperfeiçoamento de Pessoal de Nível Superior - Brazil (CAPES) - PROEX n. 0742/2020 and by the Fundação de Amparo à Pesquisa do Estado do Amazonas (FAPEAM) - PAPAC n. 005/2019. André L.S. Gonçalves received a PhD program scholarship from the FAPEAM. The author received grants from the Fundo Brasileiro para a Biodiversidade (FUNBIO) in partnership with the Instituto Humanize, the Rufford Foundation (grant number 26067-1), Columbus and Fresno Chaffee Zoo. There was no additional external funding received for this study. The funders had no role in study design, data collection and analysis, decision to publish, or preparation of the manuscript.

### Grant Disclosures

The following grant information was disclosed by the authors:
Coordenação de Aperfeiçoamento de Pessoal de Nível Superior: 0742/2020.
Fundação de Amparo à Pesquisa do Estado do Amazonas (FAPEAM): 005/2019.
Fundo Brasileiro para a Biodiversidade (FUNBIO).
Rufford Foundation: 26067-1.
Fresno Chaffee Zoo.

### Competing Interests

The authors declare that they have no competing interests.

### Author Contributions

- André L. S. Gonçalves conceived and designed the experiments, performed the experiments, analyzed the data, prepared figures and/or tables, authored or reviewed drafts of the article, and approved the final draft.
- Tadeu G. de Oliveira conceived and designed the experiments, authored or reviewed drafts of the article, and approved the final draft.
- Alexander R. Arévalo-Sandi performed the experiments, analyzed the data, authored or reviewed drafts of the article, and approved the final draft.
- Lucian V. Canto performed the experiments, analyzed the data, authored or reviewed drafts of the article, and approved the final draft.
- Tsuneaki Yabe conceived and designed the experiments, performed the experiments, authored or reviewed drafts of the article, and approved the final draft.
- Wilson R. Spironello conceived and designed the experiments, performed the experiments, authored or reviewed drafts of the article, and approved the final draft.

## Data Availability

The raw data is available in the Supplemental Files.

## Supplemental Information

Supplemental information for this article can be found online at http://dx.doi.org/10.7717/peerj.14374#supplemental-information.

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
