# Peer review of "Composition of terrestrial mammal assemblages and their habitat use in unflooded and flooded blackwater forests in the Central Amazon"

_PeerJ, doi:10.7717/peerj.14374_

## Round 0.1 · original submission · Major Revisions

Both reviewers agree that your article can be published after revision. They provided an excellent and comprehensive set of suggestions that you should use to improve the manuscript. Give special attention to statistical issues described by the reviewers (i.e., testing for spatial dependence) and find a better alignment between results and conclusion. The reviewers also found some sentences hard to follow. I believe a revision by a proficient English speaker can improve the text and the flow of your arguments.

Please ensure that ALL review and editorial comments are addressed in a response letter. Any edits or clarifications mentioned in the letter are also inserted into the revised manuscript where appropriate.

I look forward to seeing an improved version.

·

Basic reporting

The manuscript information is clearly presented, although I think some English editing may be necessary. The methods and results are well presented and the authors have provided the necessary background information. The reasoning for the predictions need to be included at the end of the introduction.

Experimental design

The methods used are robust and were well implemented. However, increasing the sampling area to other terra firme sites and sampling the terra firme concomitantly with the igapó would enhance the manuscript potential. Nevertheless, the manuscript present interesting results and the discussion needs some adjusts due these sampling limitations. (I recognize that the number of available camera traps, the costs, and the difficulty to implement this method in the field prevented the authors from applying a more complete study design. Indeed, I applaud the authors great sampling effort. They just need to adjust their discussion accordingly to these natural limitations).

Validity of the findings

Although the sampling effort is sufficient and the statistical analysis is adequate, the sampling design limited some of the conclusions of the study. I provide more detailed comments below.

Additional comments

This is a very interesting manuscript in which the authors have compared the assemblage of mammals between terra firme and adjacent igapó forests for the first time. This study is important because previous comparisons were made only between terra firme and várzea forests, and the authors have found a contrasting pattern in their study. Although I applaud the authors great sampling effort, there are limitations in this study that must be acknowledged. Firstly, the authors sampled the two environments in different seasons, which limits the conclusion that the mammals are moving from the terra firme to the igapó. So, the authors must discuss this. In addition, the fact that igapó environments become available for some months increases the availability of resources for mammals of the adjacent terra firme. This may increase the abundance of mammals in the adjacent terra firme. However, to conclude this, the authors should have sampled terra firme areas away from igapós and compare with the terra firme adjacent to igapós. Without this comparison, the importance of conserving the igapós does not receive great support (e.g. if we remove the igapós, the abundance of mammals in the terra firme will change? At least the species richness will stay the same.). So, I think the authors may adjust their conclusions in this regard. Lastly, I think the manuscript will be greatly benefited from exploring beta-diversity patterns. I provide more detailed suggestions below.


Abstract
Please explain what ‘terra firme’ is after mentioning this term.

‘This was further explored in terms of habitat use for 20 mammal species using occupancy models’ -> this is a sentence of ‘methods’ within the ‘results and discussion’ section of the abstract. I suggest removing this sentence, or if the authors think it is necessary, moving it to the ‘methods’ section

Change ‘with higher abundances’ to ‘with different abundances’, given that the authors do not specify in which habitat the abundances are higher.


Introduction
L63 - I suggest changing "habitat diversity" to "habitat characteristics" or "habitat heterogeneity".

L81-82 - I do not follow why the level of suspended sediments in the water makes the floodplain narrow. Please explain.

L93 - Note that primates are also non-volant. Maybe 'terrestrial' would be a better term, although the assemblage also includes scansorial and semi-aquatic mammals. So, non-volant and non-arboreal mammals may be a more precise term.

L119 - Although mammals really have high taxonomic resolution and high diversity, I do not agree with 'great abundance'. Compared to invertebrates, amphibians, reptiles and birds of tropical forests, mammals typically have lower abundances.

L141-147 - Please provide the reasoning for these predictions.


Study area
L153 - 'Conservation units' is a term used only in Brazil. Use 'protected area' to reach an international audience.

Although I am not a native English speaker, I had some difficulty understanding some sentences in this section and noted some odd phrasing. I feel that the text could benefit from a language review, especially in this section.



Material and Methods
L197-198 - If the cameras stayed at the sampling point for a minimum of 60 consecutive days, the mean cannot be lower than 60 days. I understand that this difference is based on the manfunctioning of some cameras, however, this needs to be better explained here.

L226 - I suggest changing "diversity measurements" to "assemblage attributes", given that abundance and biomass are not diversity measurements.

L253-255 - The authors state that they have used the iNEXT package. However, this package uses the interpolation and extrapolation method derived by Chaoet al. (2014) to build the curves, not the first order jackknife estimator. So, please clarify which method was used.

L265 - Please include the citation of the vegan package here.

L266-267 - Put the 2 in superscript


Results
L331- 4,730 trap 'days'

L336-337 - Here it is not clear if the squirrel and the two marsupial species were also removed from the analysis. The explanation for the exclusion encompasses only the primates.

Figure 2b - In the legend, interpolated and extrapolated lines look the same. This is probably due to the size of the legend. If it is not possible to correct this issue, the authors can remove this feature from the figure, since it is already described in the text legend.
Note that overlapping of the 95% confidence intervals does not guarantee that there is no statistical difference. Please see these references:
Chao, A., Gotelli, N. J., Hsieh, T. C., Sander, E. L., Ma, K. H., Colwell, R. K., & Ellison, A. M. (2014). Rarefaction and extrapolation with Hill numbers: a framework for sampling and estimation in species diversity studies. Ecological monographs, 84(1), 45-67.
Schenker, N., & Gentleman, J. F. (2001). On judging the significance of differences by examining the overlap between confidence intervals. The American Statistician, 55(3), 182-186.

L343-344 - Please include a dispersion measure associated with the relative biomass (e.g. range, standard deviation)

L344-348 - Does the overall evenness follow the same pattern as the evenness per camera? I think this is an interesting result that should be included in the manuscript.

L349 - correct to 'species'

L363-365 - Tayassu pecari, Priodontes maximus, Hydrochoerus hydrochaeris and Pteronura brasiliensis are shown in both forest types in Figure 3. Maybe the small bar size represents 0, but it is confusing. So, I think the species must be removed from the plots of the forest type where they do not occur.

L381 - Isn't the occupancy parameter of P. yagouaroundi 0.30?


Discussion
The reduction in the abundance and biomass is somewhat expected by the increase in the available space. I wonder whether there is a reduction in the abundance and biomass of mammals in the terra firme when some individuals move to igapó areas. It would have been an interesting investigation if some of the cameras were kept in the terra firme during the dry season. So, this reduced biomass and abundance of mammals does not necessarily reflect the low nutrient status of the Igapó forests. I think the authors must consider this in the discussion.

The results of the NMDS coupled with the similar species richness between terra firme and igapó forests, evidence that the beta-diversity is greater in the igapó. I think the authors could explore this aspect of the diversity and provide beta-diversity results. In line 508, the authors relate the assemblage differences to the abundance of some functional groups, and not to species turnover, however, they could provide formal beta-diversity partitioning results to support this (see Carvalho et al. (2012). Determining the relative roles of species replacement and species richness differences in generating beta‐diversity patterns. Global Ecology and Biogeography). The authors can also explore the reason for this greater beta-diversity in igapó forests. The paragraph from L468-478 relates this pattern to habitat heterogeneity and links this to the water dynamics. However, it seems that the water level did not change during the sampling of igapós, right? So, animal distribution may not be directly related to the water level, although it may be indirectly related to water level through vegetation structure, for example. Another aspect that may affect beta-diversity is that the igapó sampling points are spread over a much wider area. Is the distance between sampling points related to the beta-diversity among them?

Although I agree that the results provide some evidence for the lateral movement hypothesis, this hypothesis does not receive full support from the study design. To achieve this, the author should provide evidence that the animals occupying the igapó forests came from the adjacent terra firme forests. This may be evident for some species that cannot use the flooded environment. However, this is not so clear for scansorial species, such as Didelphis, Nasua, Eira, Tamandua, among others, and even for P. onca, which is capable of climbing onto trees and swimming.

Conclusion
L 512-514 - The authors cannot conclude this since there is no sampling of the assemblage of the igapó during the flood and of the terra firme during the dry season, and there is no sampling of other possible drivers of mammal distribution through these environments. It is clear that the flood is important, however, the authors cannot conclude that this is the most important factor.

Reviewer 2 ·

Basic reporting

I thank for the opportunity of reading this manuscript and commend the authors for this nice piece of work. The authors present an interesting study comparing terrestrial mammal assemblages in different forest types in the Amazon, showing that although they presented similar species richness, the assemblages differ in their abundance, biomass, and evenness. The manuscript is well written, and to my knowledge, data analysis and statistics are adequate.

Experimental design

I have only two concerns:
1. The first is about the spatial independence between trappings stations in terra firme and igapó, particularly those on the borders between forest types. Some of the stations are (apparently) overlapping or very close, considering Figure 1C. Could not these stations closer to terra firme represent a bias concerning species richness? i.e., they present higher richness than other stations located further within igapó? If you perform a spatial correlation test, the closest stations between forest types would likely be correlated;
2. Based on my ignorance of not knowing the studied systems, the change between forest/vegetation types is abrupt or smooth? Looking at your map, the difference seems to be mainly related to variation in elevation. I ask because this could be a confusion factor. For example, within igapó, do stations in lower elevations (i.e., possibly more affected by flooding) present fewer species, and lower abundance/biomass, than stations in higher elevations? Did you consider including in your occupation/detection models elevation and slope, or terrain roughness? Maybe these variables would make species patterns clearer.

Validity of the findings

no comment

Additional comments

I have also made a few minor comments that I hope will help the authors to improve their manuscript quality.

Introduction
I suggest changing from ‘non-volant mammals’ to ‘terrestrial mammals’ because non-volant mammals encompass species with several locomotor habits, including those with arboreal and scansorial habits, which occupation may be favored in flooded areas. Conversely, terrestrial species should be the most affected by the variation in flood regimes. In fact, you use both terms throughout the text, thus, I recommend using only terrestrial mammals, also changing it in the title.

Results
Figure 4 – I suggest including the name of the species in the graphic for the readers not familiar with Neotropical mammals.

Discussion
L 463-467 – Would not Leopardus wiedii also thrive in igapó because it is a scansorial species, i.e., with a locomotor habit slightly different from the other felids, which are more terrestrial?

---

## Round 0.2 · Minor Revisions

Please address all additional comments by reviewer one because they will improve your paper. I look forward to seeing an updated version soon.

·

Basic reporting

The manuscript is well-written. The authors have provided enough information to support their results and conclusions. The literature review is also sufficient.

Experimental design

The research question is original and relevant. No study has explored the dynamics of terrestrial mammal assemblages in igapós and adjacent terra firme forests before. The experimental design is rigorous and described with sufficient details. The authors have properly discussed the study's limitations.

Validity of the findings

The authors have carried out the necessary analyses to support their results and conclusions. The statistical analyses are adequate and robust.

Additional comments

The authors did an excellent job in addressing my comments on the first version of this manuscript. Indeed, the manuscript has improved considerably, although there is a few minor points that need to be addressed before the manuscript is suitable for publication. I detail these points below:

L150 - igapówas -> a space is missing here

L152-153 - The reasoning of this prediction should be better explained. Considering the supply of shoots and seeds in the igapó, one should expect that some species should move from terra firme to igapó to consume these items. Then, these species should contribute to a greater similarity (not dissimilarity) because they are both in terra firme and igapó, right?

L174 - height.In -> a space is missing here

L260 - Please include a reference that explains the use of Hill numbers after 'q=0'.

L297 - Please correct the ‘beta’ in 'Bbal'.

L299 - Please remove the parentheses from 'Bgra'

L378-381 - I still think it is important to present the overall evenness of both environments (i.e. the evenness considering all the records together). The authors present only the average evenness of the camera traps, which may be different from the overall evenness, considering that there is also come variation in composition between the camera traps.

L409-414 - It is great that the authors have included beta diversity analyses in their manuscript, and the approach of Basega (2017), which considers abundance changes, is even better than the one of Carvalho et al. (2012). However, the main point of the manuscript is the comparison of the assemblages between terra firme and igapó. Thus, I think the authors should present the beta diversity patterns comparing these two environments.

L460-469 - As I pointed out in the previous version of the manuscript, the reduction in biomass and abundance in igapós may not be linked only to the lower productivity of these environments. Considering that some animals are moving from terra firme to the igapó, we expect that there will be a reduction in the biomass and abundance in both environments, given that animals are now spread over a greater area. This does not mean that productivity has not an effect here, however, there are other possible explanations that must also be considered. So, I think this should be discussed by the authors.

Reviewer 2 ·

Basic reporting

I commend the authors for addressing my and the other reviewer’s comments. I am satisfied with the current version of the manuscript. Congratulation on this interesting study.

Experimental design

no comment

Validity of the findings

no comment

Additional comments

no comment

---

## Round 0.3 · accepted · Accept

Congratulations! Please work with our production team to get your paper published.